# Potent antibody lineage against malaria transmission elicited by human vaccination with Pfs25

Brandon McLeod[1,2], Kazutoyo Miura [3], Stephen W. Scally[1], Alexandre Bosch[1], Ngan Nguyen[4], Hanjun Shin[4], Dongkyoon Kim[4], Wayne Volkmuth[4], Sebastian Rämisch [5], Jessica A. Chichester[6], Stephen Streatfield[7], Colleen Woods[8], William R. Schief [5], Daniel Emerling[4], C. Richter King [8] & Jean-Philippe Julien [1,2,9]

Transmission-blocking vaccines have the potential to be key contributors to malaria elimination. Such vaccines elicit antibodies that inhibit parasites during their development in *Anopheles* mosquitoes, thus breaking the cycle of transmission. To date, characterization of humoral responses to *Plasmodium falciparum* transmission-blocking vaccine candidate Pfs25 has largely been conducted in pre-clinical models. Here, we present molecular analyses of human antibody responses generated in a clinical trial evaluating Pfs25 vaccination. From a collection of monoclonal antibodies with transmission-blocking activity, we identify the most potent transmission-blocking antibody yet described against Pfs25; 2544. The interactions of 2544 and three other antibodies with Pfs25 are analyzed by crystallography to understand structural requirements for elicitation of human transmission-blocking responses. Our analyses provide insights into Pfs25 immunogenicity and epitope potency, and detail an affinity maturation pathway for a potent transmission-blocking antibody in humans. Our findings can be employed to guide the design of improved malaria transmission-blocking vaccines.

[1] Program in Molecular Medicine, The Hospital for Sick Children Research Institute, 686 Bay Street, Toronto, ON M5G 0A4, Canada. [2] Department of Biochemistry, University of Toronto, 1 King's College Circle, Toronto, ON M5S 1A8, Canada. [3] Laboratory of Malaria and Vector Research, National Institute of Allergy and Infectious Diseases, National Institutes of Health, 12735 Twinbrook Parkway, Rockville, MD 20852, USA. [4] Atreca, 500 Saginaw Drive, Redwood City, CA 94063-4750, USA. [5] Department of Immunology and Microbial Science, The Scripps Research Institute, La Jolla, CA 92037, USA. [6] Gene Therapy Program & Orphan Disease Center, Perelman School of Medicine, The University of Pennsylvania, Philadelphia, PA 19104, USA. [7] Fraunhofer USA Center for Molecular Biotechnology CMB, 9 Innovation Way, Newark, DE 19711, USA. [8] PATH's Malaria Vaccine Initiative, 455 Massachusetts Avenue NW Suite 1000, Washington, DC 20001, USA. [9] Department of Immunology, University of Toronto, 1 King's College Circle, Toronto, ON M5S 1A8, Canada. Correspondence and requests for materials should be addressed to J.-P.J. (email: jean-philippe.julien@sickkids.ca)

Malaria remains a critical problem for human global health, exemplified by the estimated 219 million cases and approximately 435,000 deaths in 2017[1]. More than 90% of deaths related to malaria infection occur in Africa, where the parasite *Plasmodium falciparum* is responsible for >99% of clinical cases[1]. *P. falciparum* is therefore the primary target of malarial prevention strategies. Despite the severe morbidity and mortality associated with the disease, a highly efficacious and durable vaccine has yet to be developed. A major obstacle to designing a malaria vaccine is the complex *P. falciparum* lifecycle, which occurs in both a human host and a female *Anopheles* mosquito vector[2–4]. As the proteomic expression profile of the parasite changes dramatically throughout its development, the identity of antigens displayed to the immune system varies greatly, and hence a combination of vaccination approaches that target its multiple life stages may be ideal[2,5,6]. The most advanced vaccine for malaria is the RTS,S/AS01 subunit vaccine approved for use in pilot implementations in parts of Ghana, Kenya, and Malawi[7,8]. RTS,S is designed to elicit antibodies and T cells that block infection of the human liver following a mosquito bite. A complement to this strategy is to block infection at a different stage of *P. falciparum* development, the sexual stage in the mosquito, to prevent subsequent transmission of malaria-causing parasites to humans[9–11]. Vaccines designed to induce antibodies that function in the mosquito to prevent transmission to humans have been termed transmission-blocking vaccines (TBVs).

One of the most attractive TBV targets is Pfs25, a glycophosphotidylinositol-linked protein expressed on the surface of ookinetes, which develop from the fertilized zygote[12,13]. Pfs25 is important for ookinete survival in the protease-rich mosquito midgut, and Pfs25 assists in the invasion of the midgut epithelium and maturation of the oocyst[14,15]. Pfs25 exhibits minimal sequence variation between isolates, presumably not only due to functional constraints but also likely due to the fact that Pfs25 is expressed solely inside the mosquito and does not undergo the selective pressures of the human immune system[16,17]. When present in a blood meal, anti-Pfs25 antibodies significantly inhibit the development of oocysts in an in vivo standard membrane feeding assay (SMFA)[12,18,19]. Together, these data suggest that, if anti-Pfs25 antibodies are present in humans before a mosquito feeds on parasite-infected blood, *P. falciparum* development and onward transmission can be inhibited.

The main challenge in developing effective TBVs is to elicit highly potent antibodies at sufficient titers for durable transmission-blocking activity. To increase the magnitude of humoral responses, multimerization of Pfs25 and its combination with different adjuvants have been explored. In comparison to soluble monomeric protein, Pfs25 has been shown to be substantially more immunogenic when presented in a multimeric format across different particle platforms, including liposomes and viral particles[20–22]. Similarly, an increase in anti-Pfs25 titers and functional antibody responses has been observed when Pfs25 is conjugated with bacterial ExoProtein A in animal models[23]. Limited clinical testing of Pfs25 vaccine candidates have been conducted to date, and in those cases only modest levels of functional transmission-blocking activity was observed[24–26]. Until recently, there has been very little information available to understand the level of functional activity of induced human antibody responses. To begin to bridge this gap, we reported on the structure and function of monoclonal antibodies (mAbs) against Pfs25 elicited by a plant-produced Pfs25 displayed on alfalfa mosaic virus-like proteins (Pfs25-VLP) in Kymice, where the murine immunoglobulin (Ig) gene repertoire is replaced with that of humans[27,28]. Two distinct immunogenic sites were defined on Pfs25, with the most potent antibodies being directed to the epitope competing with the murine antibody 4B7, a well-characterized and potent mAb to Pfs25[27,29].

Here we explore the structure and function of transmission-blocking antibodies elicited in a human vaccinated with Pfs25-VLP[30]. We determine the co-crystal structures of four human antibodies bound to Pfs25, and describe three novel protective epitopes on Pfs25, including the most potent mAb yet described from any organism. Studies of the plasmablast lineage from which this potent transmission-blocking antibody derives reveal the evolutionary steps required for potent parasite inhibition. These insights into the maturation of potent transmission-blocking antibodies in a human will help guide the development of increasingly effective biomedical interventions against malaria.

## Results

**Pfs25-VLP vaccination elicits a diverse plasmablast response.** A total of 44 healthy human subjects were vaccinated in a dose-escalation study with the plant-produced Pfs25-VLP adjuvanted in Alhydrogel®[30,31]. One of the individuals with the highest transmission-reducing activity (TRA) determined by SMFA was chosen for further cellular characterization[30] (Supplementary Fig. 1). Plasmablasts were isolated and sequenced to obtain paired heavy- and light-chain genes as previously reported[32,33]. Representative mAbs from each of the most expanded 38 blood plasmablast lineages were expressed and assessed by enzyme-linked immunosorbent assay (ELISA) for binding to Pfs25-VLP, recombinant Pfs25, and VLP alone and classified according to their binding preference (Fig. 1 and Supplementary Fig. 2). Of these mAbs, 18 specifically bound the Pfs25 monomer, two bound only Pfs25-VLP, 16 bound the VLP alone, and two did not bind any probes. Anti-Pfs25 mAbs that exclusively bound the Pfs25 monomer were tested in SMFA to determine their TRA at a

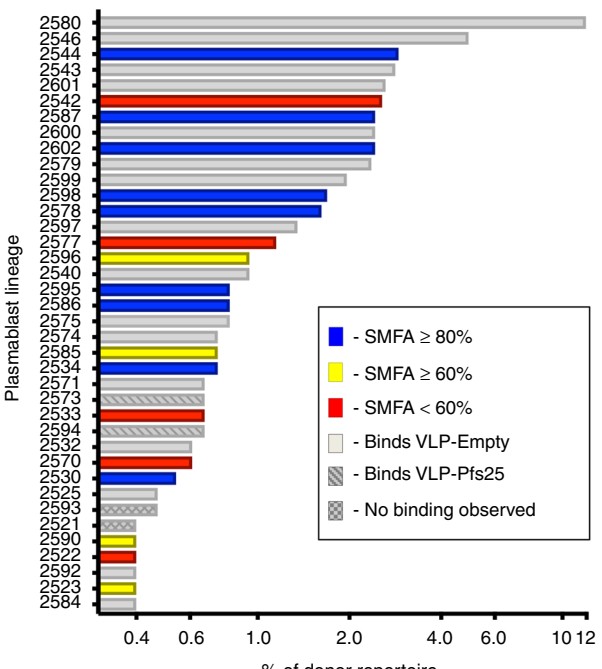

**Fig. 1** Plasmablast responses of a human donor to Pfs25-VLP immunization. Rows correspond to different antibody lineages. Labels on the left correspond to a representative lineage member that was purified, expressed, and tested for binding activity by ELISA with Pfs25 monomer, Pfs25-VLP (Pfs25+VLP) or VLP-empty (VLP alone with no antigen). VLP-empty refers to only the virus-like particle that displays the antigen, as previously described[31]. Antibodies that bound to the Pfs25 monomer were further tested by SMFA at 100 μg/mL

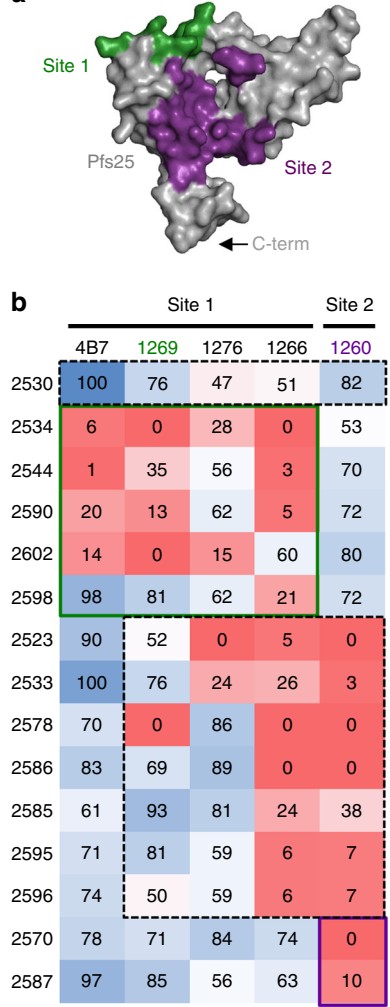

**Fig. 2** Epitope binning of human anti-Pfs25 antibodies. **a** Surface representation of Pfs25 showing the two previously characterized epitopes: Site 1 (1269, green) and Site 2 (1260, purple), and the C-terminus (C-Term) where Pfs25 was fused to the VLP. **b** Previously characterized Fabs are listed along the top and are the primary Fabs in the BLI competition assay. Secondary (competing) human Fabs are to the left. Reported scores are a percentage of total potential binding of that antibody, and therefore higher numbers (blue) display low amounts of competition, while low numbers (red) exhibit greater competition. Any experiment with >100% binding was given a score of 100, while negative values were given a score of 0. Human antibodies that fit previously defined epitopes Site 1 and Site 2 are outlined in green and purple, respectively. Potential novel epitope bins are highlighted by a dashed-line box

concentration of 100 μg/mL. Nine of these mAbs (2530, 2534, 2544, 2578, 2586, 2587, 2595, 2598, and 2602) demonstrated >80% reduction in oocyst density at 100 μg/mL concentration, while the other 9 mAbs exhibited inhibition <80% at this concentration (Fig. 1). The most expanded Pfs25-reactive plasmablast lineage was that of mAb 2544 and represented approximately 3% of the total repertoire.

**SMFA-active human mAbs bind a range of epitopes on Pfs25.** Two distinct immunogenic sites on Pfs25 had previously been defined structurally by characterizing mAbs derived from Pfs25-VLP immunization in Kymice[27]. Here we used biolayer interferometry (BLI) to conduct competitive binding experiments between human-derived Fabs and Kymab Fab 1260 (prototypic

Site 2 epitope); Kymab Fabs 1266, 1269; and 1276 (prototypic Site 1 epitopes); and the well-characterized murine 4B7 Fab (also Site 1 epitope) (Fig. 2). Of the 18 mAbs predicted by ELISA to specifically interact with Pfs25, 1 did not express in sufficient quantities as a Fab for analysis, and 2 did not bind Pfs25 with detectable affinity via BLI. Of the 15 Fabs tested, 5 Fabs (2534, 2544, 2590, 2598, and 2602) competed specifically with known antibodies directed to Site 1 or its sub-sites, and 2 Fabs (2570 and 2587) competed specifically with the Site 2-targeted antibody, corroborating the previous delineation of Pfs25 immunogenicity[27]. Interestingly, seven Fabs (2523, 2533, 2578, 2585, 2586, 2595, and 2596) competed with known antibodies from both Sites 1 and 2, suggesting that these sites may not be completely disparate but linked in one large immunogenic face, here termed the "bridging" epitope. Notably, mAb 2530 did not compete clearly with any previously defined site, suggesting mAb 2530 may bind a novel epitope on Pfs25, designated Site 3. The binding affinities of these Fabs to recombinant Pfs25 ranged between 1.7 and 181 nM, and no clear association was observed between binding strength and which epitope was targeted (Supplementary Fig. 3). mAbs 2530, 2544, 2586, and 2587 have amongst the highest affinities to recombinant Pfs25 within their respective epitope bins (Fig. 3 and Supplementary Fig. 3) and also show >80% oocyst reduction at 100 μg/mL in SMFA.

**Crystal structures delineate novel Pfs25 epitopes.** We next sought to delineate the epitopes of human mAbs, focusing primarily on specificities that had not yet been structurally characterized. Complexes of Fab 2530-Pfs25, Fab 2544-Pfs25, Fab 2586-Pfs25, and Fab 2587-Pfs25 were crystallized and diffracted to 2.0, 2.9. 3.1, and 3.1 Å resolution, respectively (Fig. 3, Supplementary Fig. 4, and Table 1). Each of the four structures corroborated the competition data, revealing novel epitopes and extended immunogenic faces beyond the previously distinct Sites 1 and 2 characterized for antibody recognition by SMFA-active antibodies elicited in Kymice[27].

2586 binds directly in between Sites 1 and 2, with HCDR3 and κCDR3 each overlapping a portion of both sites. 2586 has an epitope on Pfs25 nearly identical to 2587, despite a slightly different competition profile (Supplementary Fig. 5). This difference is explained by a slightly different angle of approach, resulting in the 2586 κCDR3 partially impeding access to Site 1. 2587 was identified as a Site 2-directed human Fab due to its strong competition with 1260. However, the crystal structure supports its categorization as a bridging antibody, also helping to explain its partial competition with Site 1-directed 1276. Notably, the only other Site 2-directed Fab, 2570, demonstrated poor TRA, even at a high concentration of 100 μg/mL, confirming the poor potency previously associated with antibodies targeting this site[27]. 2530 binds almost exclusively to the Pfs25 EGF2 domain, with only few van der Waals interactions to EGF1, helping to explain its lack of clear competition with any of the previously defined mAbs (Fig. 3). This places the 2530 epitope (Site 3) on Pfs25 nearly opposite to those of 2586 and 2587. 2544 binds near the canonical Site 1 epitope but on the opposing side of the 4B7 loop when compared to the previously defined mAb 1269 epitope (Fig. 3). This shift from the prototypical Site 1-binding site puts 2544 nearly 180° across from Site 2, an epitope not previously described. A consequence of all structural epitope mapping conducted to date is to indicate that most of the Pfs25 surface is accessible as displayed on the VLP and elicits B cell responses after vaccination in humans (Supplementary Fig. 6).

**2544 shows the most potent TRA.** To determine the TRA of these antibodies against new epitope regions on Pfs25 in more

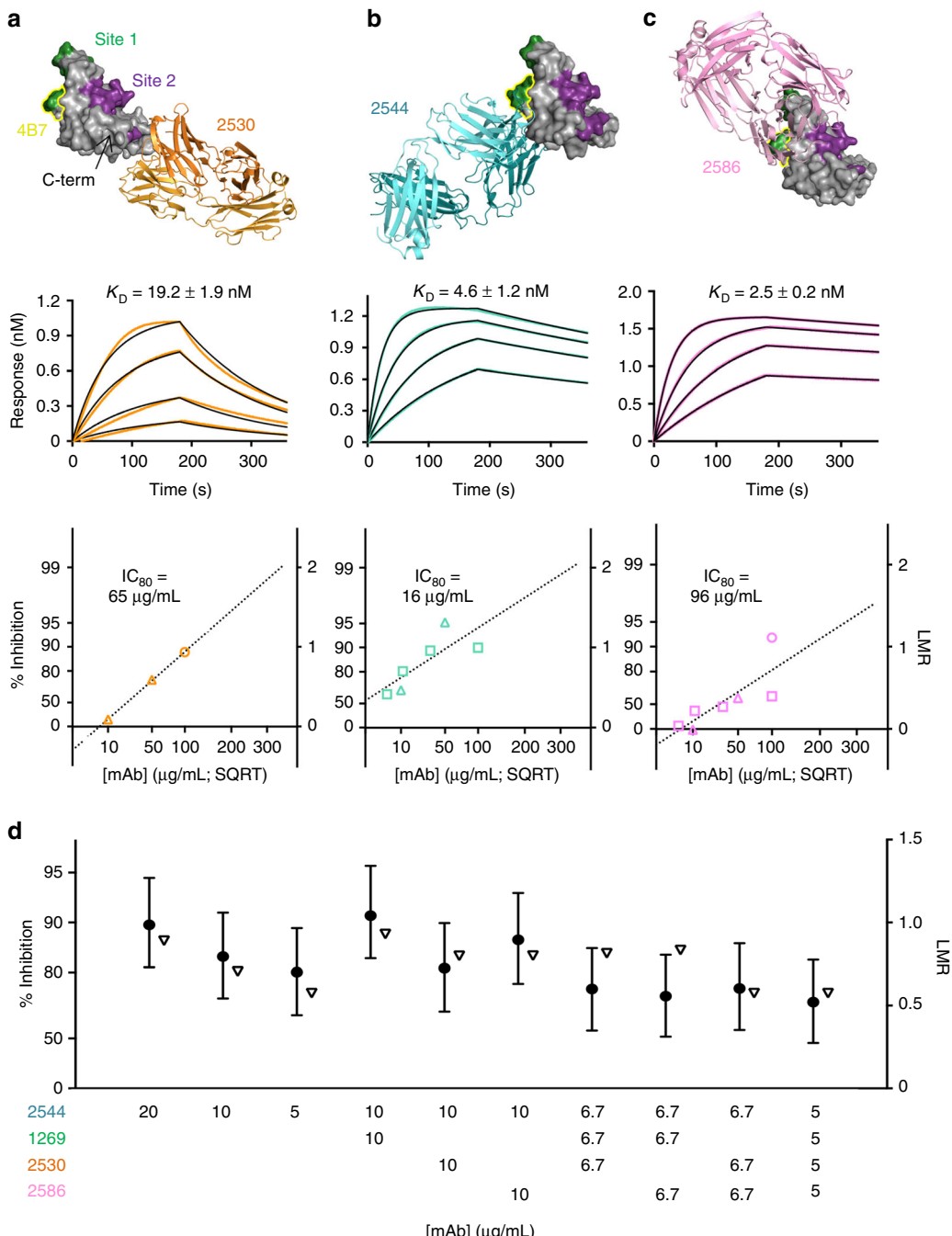

**Fig. 3** Structure–function characterization of human antibodies against Pfs25. Top: crystal structure of Pfs25 in complex with 2530 Fab (**a**, orange), 2544 Fab (**b**, teal), and 2586 Fab (**c**, pink). Site 1 (green), Site 2 (purple), the C terminus, and the 4B7 loop (yellow outline) are denoted in **a**. Middle: representative kinetic curves of 2530 Fab (**a**), 2544 Fab (**b**), and 2586 Fab (**c**) binding to Pfs25, as measured by BLI, showing association and dissociation when Pfs25 is immobilized on Ni-NTA biosensors and associated with serially diluted Fab (500–15.6 nM concentrations). The $K_D$ is indicated and derives from three replicates with error defined as standard deviation. Bottom: SMFA of 2530 IgG (**a**), 2544 IgG (**b**), and 2586 IgG (**c**) at various concentrations across two feeds. Percentage of inhibition is plotted along the left $y$-axis, log of mean oocyst reduction (LMR) is plotted along the right $y$-axis. Shapes represent different SMFA feeds, and reported $IC_{80}$ is calculated from the plotted line of best fit. **d** Combinatorial SMFA, plotted as above. Concentrations of the mAbs used are depicted along $x$-axis, and all combinations tested were at a cumulative mAb concentration of 20 μg/mL. Circles depict the best estimate of SMFA inhibition for each mAb from two feeding experiments, while upside-down triangles refer to expected inhibition. For combinatorial SMFA, expected inhibition was calculated using the Bliss independence model. Error bars are 95% confidence intervals

detail, IgGs were tested in SMFA at multiple concentrations between 5 and 100 μg/mL across multiple feeds, and $IC_{80}$ values were calculated. The $IC_{80}$ values and their 95% confidence intervals (CIs) for 2530, 2544, and 2586 were determined to be 65 [95% CI, 53–82]), 16 [95% CI, 2–41]), and 96 [95% CI, 57–145]) μg/mL, respectively (Fig. 3). Notably, 2544 has significantly higher TRA than any other mAb directed against Pfs25 yet reported, including the previously described 1269 Site 1-directed

**Table 1 Crystallographic statistics**

|  | 2530-Pfs25 | 2544-Pfs25 | 2586-Pfs25 | 2587-Pfs25 | gl2544 Fab | 2544 Fab |
|---|---|---|---|---|---|---|
| Wavelength (Å) | 0.97949 | 0.97949 | 0.97949 | 0.97949 | 0.97949 | 0.97949 |
| Space group | $P3_1$ | $P3_1 2 1$ | $P4_1 2_1 2$ | $P2 2_1 2_1$ | $C 1 2 1$ | $C 1 2 1$ |
| Cell dimensions |  |  |  |  |  |  |
| $a, b, c$ (Å) | 114.6, 114.6, 98.2 | 141.2, 141.2, 164.4 | 156.5, 156.5, 85.7 | 85.2, 110.3, 147.1 | 130.8, 60.6, 61.1 | 172.5, 43.7, 138.6 |
| $\alpha, \beta, \gamma$ (°) | 90, 90, 120 | 90, 90, 120 | 90, 90, 90 | 90, 90, 90 | 90, 113.6, 90 | 90, 111.2, 90 |
| Resolution (Å) | 40-2.0 (2.1-2.0) | 40-2.9 (3.0-2.9) | 40-3.1 (3.2-3.1) | 40-3.1 (3.2-3.1) | 40-2.0 (2.1-2.0) | 40-2.4 (2.5-2.4) |
| No. of molecules in ASU | 2 | 2 | 1 | 2 | 1 | 2 |
| No. of unique observations | 96,360 (13,044) | 42,529 (4085) | 19,905 (1770) | 25,850 (2305) | 29,727 (4002) | 38,316 (4305) |
| Multiplicity | 4.1 (4.0) | 9.9 (10.4) | 13.1 (12.7) | 6.6 (6.5) | 6.7 (6.7) | 6.7 (6.9) |
| $R_{merge}$ (%) | 10.6 (51.2) | 23.6 (83.2) | 29.3 (79.2) | 28.4 (78.4) | 8.2 (59.9) | 13.1 (72.4) |
| $R_{pim}$ (%) | 5.9 (28.8) | 8.8 (24.9) | 8.4 (23.0) | 11.9 (33.4) | 3.4 (24.7) | 5.5 (29.8) |
| $<I/\sigma I>$ | 9.5 (2.2) | 9.9 (1.4) | 8.3 (1.7) | 6.8 (1.4) | 13.6 (1.9) | 11.2 (1.6) |
| $CC_{1/2}$ | 99.7 (75.9) | 99.3 (53.3) | 98.6 (69.8) | 97.9 (54.3) | 99.9 (77.6) | 99.6 (54.9) |
| Completeness (%) | 98.8 (98.2) | 99.9 (100) | 99.9 (100) | 99.9 (100) | 99.9 (100) | 99.9 (100) |
| Refinement statistics |  |  |  |  |  |  |
| Non-hydrogen atoms | 10,194 | 9002 | 4483 | 8768 | 3442 | 6827 |
| Macromolecule | 9073 | 8978 | 4483 | 8768 | 3239 | 6667 |
| Water | 1057 | — | — | — | 197 | 124 |
| Hetero | 64 | 24 | — | — | 6 | 36 |
| $R_{factor}/R_{free}$ | 18.1/21.1 | 20.5/23.8 | 20.5/23.1 | 24.1/26.5 | 18.6/23.6 | 21.3/24.7 |
| Rms deviations |  |  |  |  |  |  |
| Bond lengths (Å) | 0.002 | 0.002 | 0.002 | 0.002 | 0.002 | 0.003 |
| Bond angle (°) | 1.18 | 0.58 | 0.51 | 0.54 | 0.58 | 0.63 |
| Ramachandran plot |  |  |  |  |  |  |
| Favored regions (%) | 96.2 | 96.5 | 96.4 | 96.1 | 98.1 | 97.7 |
| Allowed regions (%) | 3.8 | 3.5 | 3.6 | 3.8 | 1.9 | 2.3 |
| Outliers (%) | 0 | 0 | 0 | 0.1 | 0 | 0 |
| $B$-factors (Å$^2$) |  |  |  |  |  |  |
| Average | 35.2 | 68.0 | 55.9 | 61.8 | 48.0 | 70.6 |
| Macromolecule | 34.4 | 67.9 | 55.9 | 61.8 | 48.1 | 71.0 |
| Water | 40.5 | — | — | — | 46.9 | 49.3 |
| Hetero | 57.2 | 81.6 | — | — | 45.9 | 75.4 |

Highest resolution shell is given in parentheses

mAb derived from a Kymouse ($IC_{80} = 63 \, \mu g/mL$ [95% CI, 53–75]) and the murine 4B7 ($IC_{80} = 29 \, \mu g/mL$ [95% CI, 24–34])[27].

Our structural and competition analyses suggested that multiple antibodies could target Pfs25 simultaneously. We confirmed by sequential binding using BLI that recombinant Pfs25 can be bound concurrently by mAbs 2544, 1269, 2530, and 2586, with each Fab binding a different epitope (Supplementary Fig. 7). To determine how the TRA of human mAbs is affected when used in combination, SMFA experiments were conducted at a 20 μg/mL total antibody concentration, using either 2544 alone or in combination with up to three other mAbs. No synergy was observed, even when four separate epitopes were targeted simultaneously. Additive effects were observed when combining 1269 and 2544. However, owing to the substantially higher potency of 2544 compared to any other mAb, we note that higher SMFA activity is obtained with higher doses (20 or 10 μg/mL) of 2544 (Fig. 3d).

**2544 forms extensive interactions with Pfs25**. To gain understanding of the structural basis for the high potency of 2544 in SMFA, we carried out a detailed analysis of the interactions of 2544 with Pfs25. The 2544 Fab-Pfs25 crystal structure revealed that the antibody buries 1005 Å$^2$ of surface area, and the majority of this interface is contributed by the heavy chain (713 Å$^2$), in comparison to the light chain (292 Å$^2$) (Fig. 4a). Correspondingly, the heavy chain forms 21 hydrogen bonds and one salt bridge with Pfs25, compared to four hydrogen bonds formed by the light chain. The HCDR3 of 2544 contributes the largest buried surface area (BSA) of any CDR (368 Å$^2$), and forms seven

hydrogen bonds and one salt bridge, largely mediated by four tyrosines ($Y^{97}$, $Y^{99}$, $Y^{100D}$, $Y^{100F}$; Kabat numbering) that reach into a pocket of the EGF1 domain of Pfs25, immediately beside the 4B7 loop on EGF3 (Fig. 4b). Despite a slightly smaller BSA (241 Å$^2$), the HCDR2 forms 10 hydrogen bonds, five of which are mediated by $T^{56}$ (Fig. 4c). All interactions formed by the light chain with Pfs25 are mediated by κCDR1 and κCDR3, with four hydrogen bonds to the EGF4 domain of Pfs25 (Fig. 4d). κCDR2 plays a structural role to help shape the 2544 paratope but does not interact with Pfs25.

**Structural basis of 2544 affinity maturation**. Sixteen amino acid exchanges separate 2544 from its inferred germline precursor (gl2544). Notably, gl2544 does not bind Pfs25 with appreciable affinity up to the μM range as a Fab and only with $2.5 \pm 1.1 \, \mu M$ apparent binding avidity as an IgG; consequently, it fails to show any detectable inhibition in SMFA at a maximum tested concentration of 375 μg/mL (Supplementary Table 1). To understand the structural basis of affinity maturation, we solved the crystal structures of the unliganded 2544 Fab and unliganded gl2544 Fab at 2.4 and 2.0 Å resolution, respectively (Table 1). The crystal structure of unliganded 2544 showed high similarity to Pfs25-bound 2544 (overall root mean square deviation (RMSD) < 0.5 Å for the $F_v$). Comparison of the gl2544 structure with Pfs25-bound 2544 and unliganded 2544 revealed that the heavy chains are structurally similar, with a main-chain RMSD < 2.5 Å throughout the variable region, with the exception of the HCDR3, which displays major differences (nearly 12 Å in RMSD for certain residues) (Fig. 5a, top). The germline HCDR3 adopts an extended conformation in contrast to 2544, which bends at the loop apex

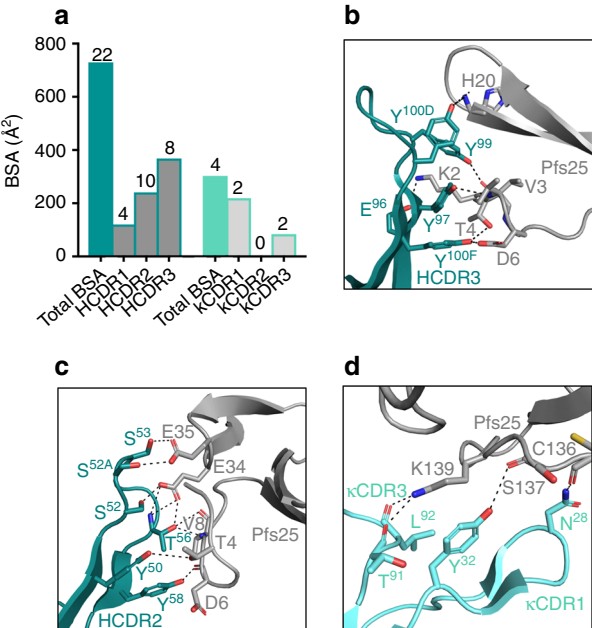

**Fig. 4** Molecular definition of the interaction between mAb 2544 and Pfs25. **a** Plot of buried surface area (BSA) of the 2544-Pfs25 complex crystal structure. BSA is separated into total interaction of heavy chain (dark teal) and light chain (light teal). Chains are further delineated into the amount of BSA per CDR, as labeled. The total number of hydrogen bonds and salt bridges per chain and CDR is listed above each bar. **b** Interaction between the 2544 HCDR3 (dark teal) and Pfs25 (gray). Both proteins are displayed as cartoons, and the side chains or main chain of residues that form hydrogen bonds are displayed as sticks. Hydrogen bonds are denoted by black dashes. **c** Interactions between 2544 HCDR2 (dark teal) and Pfs25 (gray). **d** Interactions between 2544 κCDR1/κCDR3 (light teal) and Pfs25 (gray)

(Fig. 5b). The extended conformation observed for the germline HCDR3 would be incompatible with binding to Pfs25 without conformational rearrangement.

The light-chain arrangement varies considerably more between the gl2544 and 2544 antibodies (Fig. 5a, bottom). In contrast to the light-chain N terminus that shapes the binding site of 2544, the light-chain N terminus of gl2544 adopts an elongated conformation that points away from the paratope (Fig. 5c). Notably, this striking difference occurs despite there being any residue mutation in the light-chain N terminus resulting from affinity maturation. In addition, the apex of κCDR1 is disordered in the germline structure, in contrast to 2544 where the κCDR1 is an ordered component of the binding site. From our comparative structural studies, we propose that there is an interconnectedness between the light-chain N terminus, κCDR1 and κCDR3 where somatic hypermutations in the κCDR3 result in stabilizing light-chain components in an optimal configuration for engaging Pfs25, from an otherwise flexible conformational ensemble in the germline antibody (Fig. 5c).

**High-affinity binding achieved by minimal paratope mutations.** We next investigated how the 16 somatic hypermutations in 2544 contributed to its activity. As a first step, all residues in 2544 were reverted to germline individually (except for $N^{33}S$, which would have introduced an N-linked glycosylation site not present in 2544) and binding to Pfs25 was measured by BLI. Five mutations ($G^{55}S$ and $D^{100G}Y$ in the heavy chain and $T^{91}A$, $P^{94}T$, and $F^{95}P$ in the light chain) appeared to be the most critical reversion mutations for Pfs25 binding (Fig. 5d, e). Only two of

these residues, $G^{55}$ in the heavy chain and $P^{94}$ in the light chain directly interact with Pfs25. In the unliganded gl2544 structure, $Y^{100G}$ stabilizes an extended HCDR3 conformation, which would be unfavorable for Pfs25 binding and likely explains the lower affinity for this reversion. The three mutations in the κCDR3 are near one another, and we propose these residues jointly act in stabilizing the light-chain paratope conformation to favorably interact with Pfs25.

Next, gl2544 that included the most critical 1, 3, or 5 mutations identified in the reversion experiments were expressed. The Fab construct with only one mutation from germline (κ chain $P^{95}F$; referred to as gl2544+1), showed weak but detectable binding at a maximum concentration of 1.5 μM (Fig. 5f). Since only the highest concentration showed detectable binding, a global fit to determine $K_D$ was not possible. The 3-mutation (κ chain $A^{91}T$, $T^{94}P$, and $P^{95}F$; gl2544+3) and 5-mutation Fab constructs (heavy chain $S^{55}G$ and $Y^{100G}D$; κ chain $A^{91}T$, $T^{94}P$, and $P^{95}F$; gl2544+5) bound Pfs25 with affinities of 2.0 μM and 51.1 nM, respectively (Fig. 5f). Therefore, our data indicate that it takes only three mutations from the weakly binding gl2544 to confer low micromolar affinity for Pfs25, and only five mutations to achieve nanomolar binding. The 5-mutation antibody showed a weaker-than-2544 but significant SMFA activity of 78% [95% CI, 50–90]) at 375 μg/mL (Supplementary Table 1), indicating that a gain in binding affinity to this epitope is associated with inhibitory activity, as would be expected.

**Antibodies across the 2544 lineage show potent TRA.** To better understand the development of potency for mAb 2544, we next examined the 2544 plasmablast lineage—the most expanded cellular and clonotypic Pfs25-specific lineage identified in this donor. An evolutionary phylogenetic tree of the 43 lineage members was generated through the analysis of variable domain sequences (Fig. 6a and Supplementary Fig. 8). Together, these antibodies differ from the inferred germline by 7–20 amino exchanges. Homology models were generated from all annotated antibody sequences using the 2544-Pfs25 co-complex structure. Analysis of these homology models suggested that antibodies within the lineage ranged in binding affinity (intrinsic Molecular Operating Environment (MOE) scores[34], −89 to −104), BSA (1018 to 1172 Å²), number of H bonds (10 to 19), ΔG (−8.5 to −15.7 kcal/mol), and stability (intrinsic MOE score[34], −1945 to −1979). Seven antibodies (30434, 30725, 5117, 5594, 6978, 8904, and 6012) were chosen for further characterization based on their ranking, selecting antibodies with high, intermediate, and low scores across these parameters (Table 2). Notably, 6012 possesses only seven amino acid exchanges from the inferred germline precursor, five of which are the most critical mutations for Pfs25 binding identified in the germline reversion experiments described above. Binding kinetic experiments revealed a range of affinities to recombinant Pfs25 for the selected lineage antibodies between 2.2 and 21.0 nM (Table 2). As IgGs, all selected antibodies from the 2544 lineage reduced transmission in SMFA to a similar extent and within the confidence uncertainty of the assay (Fig. 6b and Table 2). These data indicate that a ten-fold difference in binding affinity to recombinant Pfs25 in the <21.0 nM range does not significantly impact the activity of 2544-lineage antibodies within the limits of the SMFA. Furthermore, the data highlight that antibodies of the 2544 lineage retain potency even with relatively few somatic hypermutations, making this lineage an attractive target to preferentially elicit by vaccination.

**Discussion**

Our study provides the first detailed molecular analysis of the structures and potencies of human mAbs targeting Pfs25, which

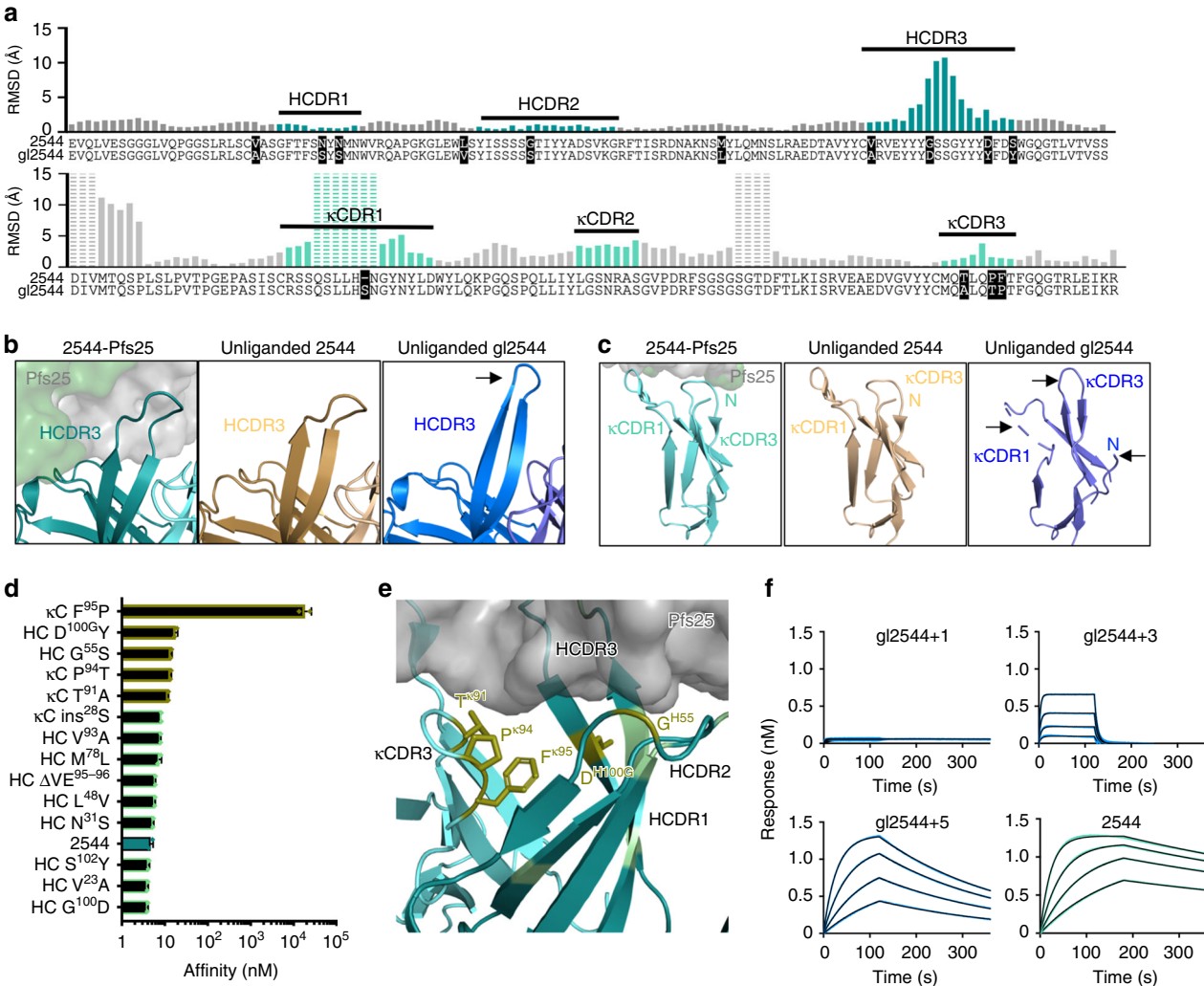

**Fig. 5** Somatic hypermutations are required for 2544 high-affinity binding. **a** Alignment and RMSD plot between the variable regions of the heavy chains (top) and light chains (bottom) of Pfs25-bound 2544 and unliganded gl2544. Somatic hypermutations from germline are highlighted in black. Gray bars of RMSD plot refer to framework regions, while heavy chain and light chain CDRs are denoted by dark and light teal bars, respectively. Dashed bars refer to residues that do not display density in at least one of the crystal structures. **b** Comparison of the HCDR3 of Pfs25-bound 2544 (teal, left), unliganded 2544 (beige, middle), and unliganded gl2544 (blue, right). Arrow indicates region of notable structural shift. **c** Comparison of the κCDR1 and κCDR3 of Pfs25-bound 2544 (teal, left), unliganded 2544 (beige, middle), and unliganded gl2544 (blue, right). Arrows indicate regions of notable structural shift. **d** Affinities of 2544 germline-reversion mutations measured by BLI (immobilized Pfs25 and mutant Fab analyte). The five reversions with the biggest impact on 2544 affinity are colored gold. Additional reversions are colored in pale green, and 2544 is colored in teal for comparison. Affinity is plotted as the mean of the measurements shown by solid diamonds, and the error is the standard deviation of the two kinetic measurements. **e** Position of the somatic hypermutations labeled in the 2544-Pfs25 crystal structure according to the scheme in **d**. Pfs25 is displayed in partial surface (gray), and 2544 is in teal. The five residues identified as most critical for binding are shown in gold sticks. **f** Representative kinetic curves for Fabs binding to Pfs25 as measured by BLI (immobilized Pfs25 and Fabs as analyte). The top left is gl2544 with the P[95]F mutation in the κ chain (gl2544+1), the top right is gl2544 with A[91]T, T[94]P, and P[95]F mutations in the κ chain (gl2544+3), the bottom left is gl2544 with A[91]T, T[94]P, and P[95]F mutations in the κ chain, and S[55]G and Y[100G]D in the heavy chain (gl2544+5), and the bottom right is 2544. A maximal Fab concentration of 1.5 μM and a minimum of 187.5 nM for the gl2544 +1 and gl2544+3 mutants, and a concentration range of 500–62.5 nM for gl2544+5 and 2544 were used for kinetic determination experiments

were elicited during a phase I clinical trial with Pfs25-VLPs[30]. Previous molecular studies of ten mAbs derived from human Ig loci transgenic mice that received the same immunogen distinguished two binding sites on Pfs25, with Site 1 being considerably more immunogenic and targeted by more potent antibodies[27]. Notably, the extent to which humanized mice and other alternative antibody-generation platforms are reflective of human responses to vaccination remains unclear, reinforcing the need to characterize antibody responses from human vaccination. The human mAbs reported here bind a broader range of epitopes, including epitopes that bridge Sites 1 and 2, and another distinct epitope termed Site 3. Notably, the Site 3-directed mAb 2530

interacts with Pfs25 in a nearly identical manner as 2A8, the canonical mAb to the P25 homolog in *Plasmodium vivax*[35]. A study in wild-type mice had suggested that this epitope was particularly immunogenic[12], yet 2530 has one of the least clonally expanded lineages of inhibitory Pfs25 antibodies and is the only Pfs25-directed mAb yet observed to this epitope. It remains unclear how the immunogenicity of Pfs25 and Pvs25 may differ, but our combined data from humanized mice and humans suggest that Pfs25 is primarily immunogenic around Site 1 (Fig. 7a).

Our analysis also allows an assessment of the relationship between antibody-binding affinity and functional potency. Within each Pfs25 epitope bin, a range of affinities and molecular

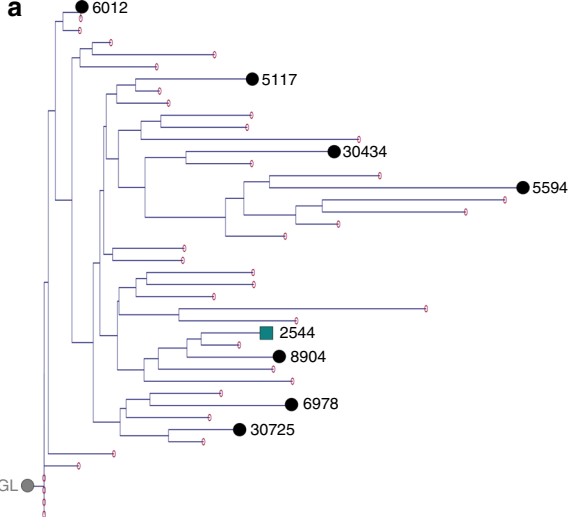

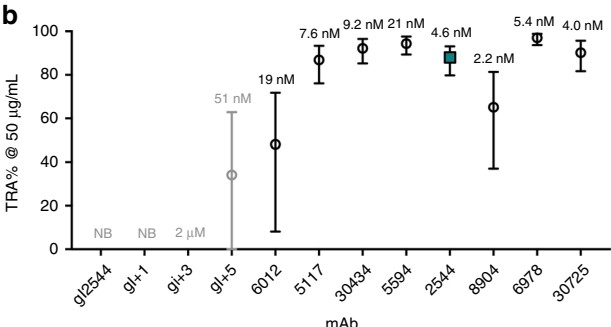

**Fig. 6** The 2544 lineage shows potent SMFA activity. **a** Phylogenetic tree of the 2544 lineage. The gray, leftmost circle refers to the position of the inferred germline precursor. Red circles indicate sequenced members of the lineage; the length of blue lines represents the evolutionary distance from germline. Antibodies chosen for further characterization are denoted by a black circle and labeled. 2544 is displayed as a teal square. **b** Transmission-reducing activity (in percentage) for gl2544 (gray), germline mutants (gray), and 2544-lineage member mAbs (black) in SMFA at 50 μg/mL. $K_D$ for each Fab binding to recombinant Pfs25 is denoted above its bar of SMFA activity. NB, no binding constant measured. Data are the result of two separate feeds, and the best-estimate (circles) and the 95% CI (error bars) of %TRA values are shown

interactions are observed, and high-affinity binding to recombinant Pfs25 does not seem to be confined to any specific epitope (Supplementary Fig. 3). Interestingly, we did not detect any relationship between mAb binding kinetics to recombinant Pfs25 and potency in SMFA. For example, mAb 2544 is significantly more potent than any other characterized anti-Pfs25 antibody ($IC_{80} = 16$ μg/mL), yet has a binding affinity to recombinant Pfs25 in the same low nanomolar range ($K_D = 4.6$ nM) as a variety of other mAbs with much lower potency (1269: $K_D = 3.7$ nM, $IC_{80} = 63$ μg/mL; 1245: $K_D = 31.0$ nM, $IC_{80} = 263$ μg/mL; 2586: $K_D = 3.3$ nM, $IC_{80} = 96$ μg/mL). In addition, molecular analysis of various members of the 2544 lineage demonstrates that differing amounts of hydrogen bonds or BSA or a log difference in binding affinity to recombinant Pfs25, do not significantly correlate with transmission-blocking activity for binding the same epitope within the detection limit. Therefore, we propose that potency is primarily determined by the epitope that is engaged (Fig. 7b and Supplementary Fig. 9) and is possibly related to the angle of approach. Indeed, 2544 recognizes the 4B7 loop at a shifted angle compared to other Site 1-directed mAbs of

known structures (e.g., 1269, which has been shown to be of intermediate potency in SMFA, $IC_{80} = 63$ μg/mL, and murine 4B7, $IC_{80} = 29$ μg/mL)[27]. Epitopes distant from Site 1 are associated with substantially lower SMFA activity (1245, $IC_{80} = 263$ μg/mL; 2586, $IC_{80} = 96$ μg/mL).

Combined, these data provide a guide for improved vaccine design based on immuno-focusing to mask immunogenic surfaces associated with poor mAb inhibitory activity and thereby direct B cell responses to sites associated with potent parasite inhibition. Such structure-based approaches have been successful in preferentially eliciting antibody responses to sites of vulnerability on the HIV-1 Env[36,37], the Respiratory Syncytial Virus F protein[38], and the influenza hemagglutinin[39,40], among other examples. Our findings that human-derived Pfs25 polyclonal antibodies do not appear to act synergistically (Fig. 3d) strongly suggest that eliciting high titers of the most potent mAb lineages will be critical to improve vaccine potency. Furthermore, with antibodies of less potency than 2544 partially competing for its epitope, it may be important to design an immunogen that preferentially elicits the most effective mode of binding to Pfs25. Indeed, despite the 2544 lineage being the most expanded Pfs25-specific plasmablast lineage in this vaccine, it only represented 3% of the total repertoire, with the rest evidently targeting unrelated antigens, the immunogen coat protein, or regions on Pfs25 associated with lower inhibitory potency. Consequently, this unfocused polyclonal response resulted in the modest TRA detected in the clinical trial at the serum level (approximately 30% at 3.5 mg/mL[30]).

Our molecular work also shows how a potent human mAb can develop with as little as seven amino acid mutations (mAb 6012) from an inferred germline B cell precursor. This suggests that vaccines targeting this (or a similar) lineage may rapidly induce the development of protective antibodies, even when minimally mutated from their germline precursor. Benefits of such an approach would include a reduced immunization schedule. The very low affinity of gl2544 for Pfs25 suggests that efficient vaccine elicitation of 2544-like antibodies might benefit from the design of a slightly modified Pfs25 antigen with appreciable affinity for 2544-like naive B cells. Such protein-engineering efforts are being extensively explored in HIV vaccine development, with technologies being developed to more efficiently engage specific germline precursors[41–44]. Combining next-generation immunogens with potent adjuvants that elicit high titers and durable responses will also be critical for vaccine efficacy, as preclinical studies have shown how different combinations of Pfs25 and adjuvants can result in vastly different antibody responses[45–47].

Finally, our identification of a highly potent human mAb against Pfs25 offers the opportunity to test its efficacy in passive immunization trials, such as is currently being developed for other transmission-blocking targets, e.g., Pfs48/45[48]. Such data would be critical to further validate Pfs25 as a TBV target and would determine the circulating antibody concentrations required to block the transmission of the parasite, thereby guiding targets for future Pfs25-based vaccine evaluation. The fine mapping of epitopes and their associated potency derived in our studies will also provide new tools to probe immune responses in future experimental animal and human studies, such as determining the proportion of a polyclonal response to Pfs25 that binds the most potent epitopes. Taken together, our data provide a molecular framework for understanding human antibody responses to Pfs25 and for designing improved transmission-blocking biomedical interventions.

## Methods

**Identification and selection of Pfs25-specific mAb sequences**. Our analysis focused on the B cell response of one participant who achieved high TRA (76.9% at

### Table 2 Characterization of mAbs from the 2544 lineage

| mAb | Mutations from gl | BSA (Å²) | HB | SB | $\Delta G$ + HB/SB (kcal/mol) | MOE affinity | MOE stability | $K_D$ (M) | TRA (%) at 50 µg/mL (95% CI) |
|---|---|---|---|---|---|---|---|---|---|
| gl2544 | — | 1097 | 12 | 1 | −10.8 | −96.7 | −1950.3 | NB | 1 (−75 to 45) |
| 2544 | 15 | 1005 | 25 | 1 | −15.6 | −102.6 | −1967.3 | $4.6 \times 10^{-9} \pm 1.2 \times 10^{-9}$ | 88 (80 to 93) |
| 30434 | 19 | 1140 | 15 | 3 | −15.2 | −98.9 | −1977.6 | $9.2 \times 10^{-9} \pm 6.1 \times 10^{-10}$ | 92 (85 to 97) |
| 30725 | 14 | 1101 | 15 | 2 | −12.5 | −102.0 | −1973.4 | $4.0 \times 10^{-9} \pm 6.4 \times 10^{-10}$ | 90 (82 to 96) |
| 5117 | 17 | 1031 | 11 | 1 | −8.5 | −94.6 | −1978.5 | $8.7 \times 10^{-9} \pm 1.6 \times 10^{-9}$ | 87 (76 to 93) |
| 5594 | 20 | 1096 | 11 | 0 | −10.4 | −95.2 | −1952.8 | $2.1 \times 10^{-8} \pm 3.5 \times 10^{-10}$ | 94 (89 to 98) |
| 6012 | 7 | 1084 | 15 | 2 | −12.6 | −95.4 | −1962.8 | $1.9 \times 10^{-8} \pm 3.7 \times 10^{-9}$ | 48 (8 to 72) |
| 6978 | 15 | 1128 | 18 | 4 | −15.7 | −95.0 | −1969.4 | $5.4 \times 10^{-9} \pm 8.6 \times 10^{-10}$ | 97 (94 to 99) |
| 8904 | 17 | 1128 | 19 | 1 | −12.4 | −101.5 | −1976.2 | $2.2 \times 10^{-9} \pm 7.5 \times 10^{-10}$ | 65 (37 to 81) |

In bold, in silico modeling

BSA buried surface area, HB hydrogen bonds, SB salt bridges, ΔG Gibbs free energy, MOE Molecular Operating Environment, NB no binding, TRA transmission reducing activity

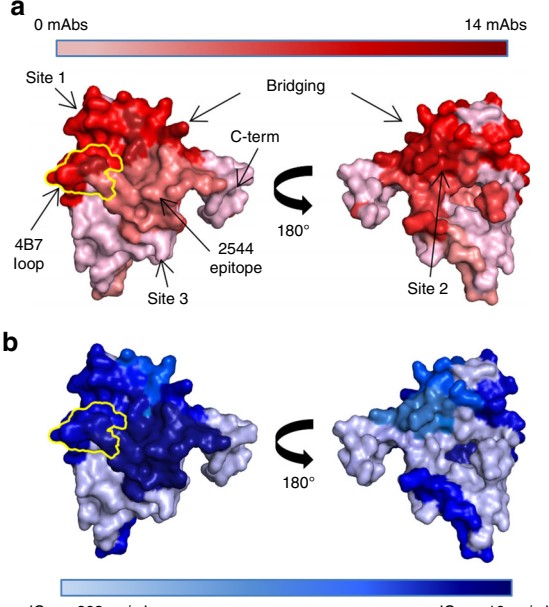

**Fig. 7** Summary of Pfs25 immunogenicity and mAb potency. **a** Binding sites on Pfs25 of mAbs isolated from a human donor and previously isolated from Kymice[27]. Pfs25 sites are delineated according to canonical binders for which crystal structures are known: 1269 for Site 1a, 2544 for Site 1b, 1260 for Site 2, 2586 for bridging, and 2530 for Site 3. Residues of Pfs25 are colored according to the quantity of mAbs expected to bind that residue from BLI competition experiments. The darker the red, the greater the immunogenicity on Pfs25. The loop on Pfs25 recognized by murine 4B7 is highlighted in yellow. **b** Surface representation of potency for anti-Pfs25 mAbs isolated from a human donor and Kymice[27]. Pfs25 sites are delineated as in **a**, and residues colored according to mAb potency ($IC_{80}$). If more than one mAb was present in an epitope bin, an average of the $IC_{80}$s was used. The darker the blue, the higher the potency of mAbs against this site in SMFA

15 mg/mL) in the phase I clinical trial study registered at http://www.ClinicalTrials.gov under reference identifier NCT02013687[30]. Plasmablasts were isolated from peripheral blood mononuclear cells (PBMCs) using flow cytometry. PBMCs were stained with the following antibodies: anti-CD19 (clone HIB19, Brilliant Violet 421, BioLegend, cat # 302234, 50 µg/mL, 1:50), anti-CD20 (2H7, PE-Cy7, BioLegend or L27, PerCP-Cy5.5, BD Bioscience, cat # 302312, 100 µg/mL, 1:100), anti-CD27 (O323, Brilliant Violet 510, BioLegend, cat # 302836, 100 µg/mL, 1:50), anti-CD38 (HIT2, Alexa647, BioLegend, cat # 303514, 200 µg/mL, 1:100), anti-IgA (IS11-8E10, fluorescein isothiocyanate (FITC), Miltenyi Biotec, cat # 130-093-071, 16.5 µg/mL, 1:200), anti-IgM (MHM-88, FITC, BioLegend, cat # 314506,

100 µg/mL, 1:100), anti-IgD (IA6-2, Alexa Fluor, BioLegend, cat # 348216, 200 µg/mL, 1:100), anti-CD3 (UCHT1, FITC, BioLegend, cat # 300406, 200 µg/mL, 1:100), and anti-CD14 (HCD14, FITC, BioLegend, cat # 325604, 400 µg/mL, 1:100). Flow cytometric analysis and cell sorting was performed on a BD FACSJazz using the BD FACS Software. Plasmablasts were isolated by gating for CD19+CD20low/−CD27+CD38highIgA−IgM−IgD−CD3−CD14− cells. All paired chain antibody sequencing was carried out on IgG plasmablasts sorted into microtiter plates at one cell per well by fluorescence-activated cell sorter (FACS) and 1484 paired chain sequences were generated using Atreca's Immune Repertoire Capture® technology as previously described[27]. A total of thirty-eight paired IgG variable regions (CDRs and framework regions) were selected for direct recombinant expression from the most expanded plasmablast lineages identified from the donor sample. This approach biased the selection toward large expanded lineages.

**Expression and purification of IgGs**. Selected IgG sequences were expressed by Lake Pharma (Belmont, CA). Proteins were expressed using 0.1 L transient production in HEK293 cells (Thermo Fisher Scientific) and purified using protein A affinity chromatography. Purified proteins were 0.2 µm sterile filtered, characterized for >90% purity by CE-SDS, concentration and endotoxin <100 EU/mg prior to vialing. Final buffer for all proteins contained 200 mM HEPES, 100 mM NaCl, and 50 mM NaOAc, pH 7.0.

**Expression and purification of Fabs**. $V_L$ and $V_H$ regions were cloned into pcDNA3.4 expression vectors upstream of the human $C_H$, Cκ, or Cλ regions. IgGs and Fabs were transiently expressed in HEK293F cells (Thermo Fisher Scientific) and purified using KappaSelect or LambdaSelect affinity (GE Healthcare), followed by cation exchange (MonoS, GE Healthcare) and size exclusion chromatography (Superdex 200 Increase 10/300 GL, GE Healthcare).

**BLI binding studies**. BLI (Octet RED96, FortéBio) experiments at 25 °C were conducted to determine the epitope bins and binding kinetics of Pfs25 and the human Fabs. Recombinant His-tagged Pfs25 purified from HEK293F cells (Thermo Fisher Scientific) through a 5-mL HisTrap FF column (GE Healthcare), followed by size exclusion chromatography (Superdex 200 Increase 10/300 GL, GE Healthcare)[27], was diluted into kinetics buffer (phosphate-buffered saline, pH 7.4, 0.01% (w/v) bovine serum albumin, 0.002% (v/v) Tween-20) at 10 µg/mL and immobilized onto Ni-NTA (NTA) biosensors (FortéBio). To determine epitope bins, baseline was gathered for 30 s before biosensors were dipped into wells containing the primary antibody (Fab, 10 µg/mL) for 10 min, followed by another 30 s baseline, and then dipped into wells containing a second antibody (Fab, 10 µg/mL) for 5 min. For binding kinetics determination, His-tagged Pfs25 was similarly diluted into kinetics buffer and immobilized onto Ni-NTA (NTA) biosensors (FortéBio). Following a 60-s baseline step, biosensors were dipped into wells containing twofold dilution series of Fab (affinity measurement) or IgG (apparent avidity measurement). Comparatively, IgG affinity was assessed by immobilizing the IgG onto anti-human Fc biosensors (FortéBio) and dipping into serially diluted concentrations of Pfs25. Sensors were then dipped back into kinetics buffer to monitor the dissociation rate. Competition and kinetics data were analyzed using the FortéBio's Data Analysis software 9.0, and kinetic curves were fitted to a 1:1 binding model using at least four concentrations. For experiments with several replicates, the mean kinetic constants reported are the result of two or more independent experiments, and associated error is standard deviation.

**Co-crystallization and structure determination**. A Pfs25 construct with potential N-linked glycosylation sites at positions 91, 144, and 166 mutated to Gln was transiently expressed in HEK293F cells (Thermo Fisher Scientific) and purified via a 5-mL HisTrap FF column (GE Healthcare), followed by size exclusion

chromatography (Superdex 200 Increase 10/300 GL, GE Healthcare), as previously described[27]. Pfs25 and human Fabs were mixed in a 2:1 molar ratio, and size exclusion chromatography was used to remove excess Pfs25. Fab–Pfs25 complexes were concentrated to approximately 10 mg/mL and mixed 1:1 with mother liquor and set up in sitting-drop crystallization experiments. 2530-Pfs25 crystals grew in 0.2 M lithium sulfate, 0.1 M Tris, pH 7.0, and 2.0 M ammonium sulfate and were cryoprotected with 15% (v/v) ethylene glycol. 2544-Pfs25 crystals were grown in 0.1 M HEPES NaOH, pH 7.5, 20% (w/v) PEG 4000, and 10% (v/v) 2-propanol and cryoprotected with 10% (v/v) glycerol. 2586-Pfs25 crystals were grown in 0.2 M lithium sulfate, 0.1 M Tris-HCl, pH 8.5, and 30% (w/v) PEG 4000. 2587-Pfs25 crystals were grown in 0.2 M potassium thiocyanate, 20% (w/v) PEG 3350, and cryoprotected in 15% (v/v) ethylene glycol. Unliganded gl2544 was crystallized in 0.1 M sodium acetate pH 4.5, 25% (w/v) PEG 3350, and cryoprotected with 15% (v/v) glycerol. Unliganded 2544 crystals were grown in 0.1 M sodium cacodylate HCl, pH 6.5, 1.0 M sodium citrate tribasic, and cryoprotected with 15% (v/v) glycerol. Data were collected at the 08ID-1 beamline at the Canadian Light Source (CLS), processed, and scaled using XDS[49]. Structures were determined by molecular replacement using Phaser and Pfs25 as a search model[27,50]. Refinement of the structures was carried out using phenix.refine and model building iterations in Coot[51,52]. SBGrid was used to access all software[53].

**Molecular modeling of 2544 lineage**. All models were generated from the 2544-Pfs25 complex crystal structure. This structure was prepared using QuickPrep in MOE, and models were generated through MOE's residue scanning function[34]. Predicted affinities between antibodies and the antigen were determined by model generation using MOE's scoring algorithms. Resulting homology models were further analyzed using the PDBePISA server[54].

**Standard membrane feeding assay**. The ability of anti-Pfs25 antibodies to reduce the development of *P. falciparum* NF54 strain oocysts in the mosquito midgut was evaluated by SMFA as described previously[55]. In brief, test antibodies at the indicated concentrations were mixed with 0.15–0.2% stage V gametocytemia and then fed to 3–6-day-old female *Anopheles stephensi* (initially provided by The Catholic University of the Netherlands). The mosquitoes were maintained for 8 days and then dissected to count the number of oocysts per midgut in 20 mosquitoes. The best estimate of %TRA, the 95% CI, and significance of inhibition from single or multiple feeds were calculated as previously described using a zero-inflated negative binomial model[56]. The Bliss independence model, i.e., assuming mAbs act independently, was used to determine the theoretical additive effect.

**Reporting summary**. Further information on research design is available in the Nature Research Reporting Summary linked to this article.

## Data availability

Crystallographic data has been deposited to the Protein Data Bank (PDB) under accession numbers 6PHB, 6PHC, 6PHD, 6PHF, 6PHG, and 6PHH. The authors declare that all other data supporting the findings of this study are available within the article and its Supplementary Information files or are available from the authors upon request.

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

## Acknowledgements
This work was funded by PATH's Malaria Vaccine Initiative under Grant OPP1108403 from the Bill & Melinda Gates Foundation. This research was undertaken, in part, thanks to funding from the Canada Research Chairs program (to J.-P.J.). S.W.S. was supported by a Hospital for Sick Children Lap-Chee Tsui postdoctoral fellowship and a Canadian Institutes of Health Research (CIHR) fellowship (FRN-396691). X-ray diffraction experiments were performed using beamline 08ID-1 at the Canadian Light Source, which is supported by the Canada Foundation for Innovation, Natural Sciences and Engineering Research Council of Canada, the University of Saskatchewan, the Government of Saskatchewan, Western Economic Diversification Canada, the National Research Council Canada, and the Canadian Institutes of Health Research. SMFA experiments were supported in part by the intramural program of the NIAID, NIH.

## Author contributions
B.M., S.W.S., D.E., C.R.K. and J.-P.J. designed experiments; J.A.C. and S.S. provided clinical samples; B.M., K.M., S.W.S and A.B. performed experiments; B.M., K.M., S.W.S., A.B., N.N., H.S., D.K., W.V., S.R., C.W., W.R.S., D.E., C.R.K. and J.-P.J. analyzed the data; B.M., C.R.K. and J.-P.J. wrote the manuscript; and C.R.K. and J.-P.J. conceived the study.

## Additional information

**Competing interests:** The authors declare no competing interests.

