## [Peer Review File · Nature Communications]

Reviewers' Comments:

Reviewer #1:

Remarks to the Author:

In this paper, McLeod et al. describe the discoveries of potent malaria-transmission blocking antibodies from a clinical trial evaluating Pfs25-VLP, determine their structures in complex with their epitopes and study the lineage of the most potent one. They conclude that there appears to be a site on Pfs25 that induces more potent antibodies than other sites, and that following the HIV vaccine field, this information can be used for vaccine design.

The paper is well written, and no major issues were raised. However, the authors can do minor revisions to clarify some points. Additionally, the importance of certain lineages and potent sites of vulnerability could be toned down as this is just an n=1, and the overall vaccine efficacy was not very high.

The data show that a polyclonal response is mounted by the immune system, which is the case in most vaccinations.

The results obtained from this study indicate that the protein is exposed to the immune system and induced polyclonal responses that target multiple sites. Maybe that is a good thing. Until it has been shown, it is unclear if there are immunodominant epitopes that are not protective (as is the case for HIV) and thus, although of interest, it is unclear if design optimization would be needed rather than adjuvant formulation as pointed out in ref 30. Maybe the authors can clarify this as a possibility.

The first two-sentences of the abstract reads "Transmission-blocking vaccines have the potential to be key contributors to malaria elimination. Such vaccines elicit antibodies that inhibit parasites as they transit from humans to Anopheles mosquitoes, thus breaking the cycle of transmission." That does not seem to agree with line 69-72 of Introduction: "A complement to this strategy is to block infection at a different stage of Pf development, the sexual stage in the mosquito, to prevent subsequent transmission of malaria-causing parasites to humans" 9-11. Vaccines designed to induce antibodies that function in the mosquito to prevent transmission to humans have been termed transmission-blocking vaccines (TBV)." Maybe the authors can clarify this. Since the vaccine is given to humans, likely it is to prevent transmission from human to mosquitoes.

The conclusion of ref30 is that "Although the trial did not meet the TRA criterion in the two higher dose groups, the 100 µg dose did demonstrate a weak, but significant, TRA after the 3rd vaccination, suggesting the potential of this vaccine to induce functional antibody titers in humans." The authors can maybe be more specific in explaining which individual they chose by showing a supplemental figure.

Line 113 - add more details here, how were plasmablasts isolated.

Figure 1, advise to thin the gray line around boxes for visualization.

Line 132 - state briefly why 3 Abs were not tested (low expression?)

Line 144 - would it be useful to see if there is a correlation between affinities and oocyst reduction in SMFA? Apparently there is not.

Line 169 - a summary figure showing the different Mabs binding to their epitopes with their associated potency in SMFA will be nice.

Line 243 - binding barely visible on the figure

Figure 5 - will be nice to see germline gene VDJ and V-J as well in the alignment

Line 282 maybe change "to" to "from" human vaccination

Discussion needs to be reviewed, the authors refer to HIV field, a lot has been hypothesized but not shown for HIV.

Reviewer #2:

Remarks to the Author:

This work expands our repertoire of human transmission blocking mAb as well as our understanding of the structure of the transmission-blocking epitopes present on Pfs25. The comparison with the germline mAb provides insight into the mutations required for high affinity binding that have the potential to inform the design of antigens that could be used sequentially to enhance the immune response. Although an attractive hypothesis for vaccine development, much further work is needed to test whether sequential antigens or another approach to focus the immune response toward specific epitopes will be sufficient to generate a robust transmission blocking response in humans. The lack of synergy between mAb that bound different epitopes was disappointing but important to know. Another challenge is the presence of antibodies that bind to the same region compete with one another yet have different transmission blocking potential. This issue should be included in the discussion.

Prior to publication I think it is important to include additional details about the plasmablast sequence analysis. How many sequences were determined? It would also be good to include a phylogram of all the sequences with the selected Ig sequences (both those that did and did not bind Pfs25) and the location of the 2544 expansion indicated. This data would give a better overview of the response. More detailed description of the structure of Pfs25 in relation to the EGF domains and the location of the gpi tail to indicate the potential orientation membrane as well as the proposed tiled alignment of Pfs25.

L114) They mention producing 38 Ig selected from most expanded lineages and then 15 fabs, but the methods indicates 96 sequences were selected for recombinant production and 75 were from the expanded lineages and the rest were highly mutated.

Fig 2B) Please include the relative TRA and binding affinity.

L153-169) Please add the location of the GPI tail to indicate the orientation with respect to the membrane and include the reference to the EGF domains targeted for all the Ig, since these are how the Pfs25 structure has been described historically.

L177) Please include the TRA of mAb 4B7 as a comparator in this section.

Add the direct comparison of the VH and VL sequences of the 2544 Ig tested for transmission blocking activity in supplemental material using the format in Fig 5A.

Reviewer #3:

Remarks to the Author:

This is a paper about trying to create a transmission blocking vaccine, something that is theoretically interesting but might be challenging in real life in humans. The authors have before published results from humanized mice, and have now moved on to humans which is very interesting, especially since the number of papers in the literature concerning trials in humans are not that big compared to the number of studies performed in mice.

To use only plasmablast analysis limits the interpretation of the studies, especially from a transmission blocking point of view where a long term response is needed to be efficient, but it is a good start towards understanding the basics of the immune response. Another limitation is that the plasmablasts have to survive for 7 days in culture, something that might in itself mean a selection bias.

Some antibodies seem to partially impede access to important sites, hence it might be difficult to predict the total outcome of Pfs25 as a vaccine when as many as 4 EGF domains are included. EGF

domains contain several S-S bridges and often fold well and form a relatively stable structure on their own, why have the authors not used a smaller protein with fewer EGF domains, why use 4 EGFs? To avoid forming (interfering) antibodies against too many sites, fewer EGF domains could potentially be used in a vaccine. It would therefore be good to include one figure to clarify which EGF domains are part of the different binding sites (site 1, site 2 etc) to facilitate an interpretation of the importance of the different domains. It would also be good to test the mAb 2544 against the single EGF domains.

How do the authors explain that it is the heavy chain that seems to contribute mostly to the interface of the surface area of 2544 in the binding to Pfs25? Which subclass of IgG is it, does it make a difference?

The authors state that they did not detect any relationship between mAb affinity to recombinant Pfs25 and potency in SMFA, and there was no association between binding strength and which epitope was targeted. Instead of looking only at KD values, was there any association with off-rates? They seem to differ more than on-rates, and off-rates should be important since if the antibodies only stay bound for a few seconds, they will not have time to confer any blocking effects.

The concentration of mAb needs to be in the range of 100µg/mL to have an effect on oocyst density. Is it realistic to think that this concentration can be achieved in human plasma? Are there other successful vaccines that have managed to reach this concentration of specific antibody? The most expanded Pfs25-reactive plasmablast lineage was that of mAb 2544, and represented approximately 3% of the total repertoire, indicating that also other (potentially interfering) antibodies will be produced and that it will be difficult to reach a high concentration of the wanted antibody. What is the actual concentration of the equivalent of 2544 in the human plasma, is it at all possible to measure it or detect it? If so, how long does it stay detectable in plasma? Since this paper has used only plasmablasts, it is of special interest to consider how long an immune response can be detected. In the discussion about the number of mutations from a germline B cell precursor, it would be of interest not only to note the number of amino acid mutations but also how big a structural change every mutation means. Some amino acids are very similar in overall structure/side chains and a change for another amino acid might not mean a big change in structure.

Reviewers' comments:

Reviewer #1 (Remarks to the Author):

In this paper, McLeod et al. describe the discoveries of potent malaria-transmission blocking antibodies from a clinical trial evaluating Pfs25-VLP, determine their structures in complex with their epitopes and study the lineage of the most potent one. They conclude that there appears to be a site on Pfs25 that induces more potent antibodies than other sites, and that following the HIV vaccine field, this information can be used for vaccine design. The paper is well written, and no major issues were raised. However, the authors can do minor revisions to clarify some points. Additionally, the importance of certain lineages and potent sites of vulnerability could be toned down as this is just an n=1, and the overall vaccine efficacy was not very high.

We thank the Reviewer for the positive assessment of our work. We have addressed the minor revisions as detailed below.

The data show that a polyclonal response is mounted by the immune system, which is the case in most vaccinations. The results obtained from this study indicate that the protein is exposed to the immune system and induced polyclonal responses that target multiple sites. Maybe that is a good thing.

It is demonstrated in this study that antibody potency correlates most directly with the epitope of Pfs25 that is engaged, and that no synergy is observed between the various epitopes (Fig. 3D). Additionally, the clinical study from which these antibodies were isolated tested the donor's polyclonal sera, which had low potency in SMFA, despite containing very potent monoclonal antibodies such as 2544 and its lineage. Therefore, we propose that an untargeted polyclonal response to the entirety of Pfs25 is a poor outcome, and that instead a polyclonal response that focuses on the most potent epitopes may lead to superior transmission-blocking activity.

Until it has been shown, it is unclear if there are immunodominant epitopes that are not protective (as is the case for HIV) and thus, although of interest, it is unclear if design optimization would be needed rather than adjuvant formulation as pointed out in ref 30. Maybe the authors can clarify this as a possibility.

We agree with the reviewer that adjuvants for Pfs25 in humans is a topic that has not been thoroughly explored, and may have a major impact to vaccine sustainability and efficacy. Hence, we have added to the Discussion (L 336-339) stating, "Combining next-generation immunogens with potent adjuvants that elicit high-titers and durable responses will also be critical for vaccine efficacy, as pre-clinical studies have shown how different combinations of Pfs25 and adjuvants can result in vastly different antibody responses⁴⁵⁻⁴⁷."

The first two-sentences of the abstract reads “Transmission-blocking vaccines have the potential to be key contributors to malaria elimination. Such vaccines elicit antibodies that inhibit parasites as they transit from humans to *Anopheles* mosquitoes, thus breaking the cycle of transmission.” That does not seem to agree with line 69-72 of Introduction: “ A complement to this strategy is to block infection at a different stage of Pf development, the sexual stage in the mosquito, to prevent subsequent transmission of malaria-causing parasites to humans^{9– 11}. Vaccines designed to induce antibodies that function in the mosquito to prevent transmission to humans have been termed transmission-blocking vaccines (TBV).” Maybe the authors can clarify this. Since the vaccine is given to humans, likely it is to prevent transmission from human to mosquitoes.

We agree with the reviewer that the wording of the abstract and introduction was confusing. In the context of Pfs25, antibodies elicited by TBVs function in the mosquito, preventing oocyst development. The abstract has been updated as follows to more clearly reflect this: “Transmission-blocking vaccines have the potential to be key contributors to malaria elimination. Such vaccines elicit antibodies that inhibit parasites during their development in *Anopheles* mosquitoes, thus breaking the cycle of transmission.” (L30-32)

The conclusion of ref30 is that “Although the trial did not meet the TRA criterion in the two higher dose groups, the 100 µg dose did demonstrate a weak, but significant, TRA after the 3rd vaccination, suggesting the potential of this vaccine to induce functional antibody titers in humans.” The authors can maybe be more specific in explaining which individual they chose by showing a supplemental figure.

As requested by the Reviewer, we have now added a Supplementary Figure (Fig. S1) to highlight which individual from the clinical trial was evaluated in the present study.

Line 113 - add more details here, how were plasmablasts isolated.

We added details in the Methods section of the manuscript to describe how plasmablasts were isolated: “Plasmablasts were isolated from peripheral blood mononuclear cells (PBMCs) using flow cytometry. PBMCs were stained with the following antibodies: anti-CD19 (clone HIB19, Brilliant Violet 421, BioLegend), anti-CD20 (2H7, PE-Cy7, BioLegend or L27, PerCP-Cy5.5, BD Bioscience), anti-CD27 (O323, Brilliant Violet 510, BioLegend), anti-CD38 (HIT2, Alexa647, BioLegend), anti-IgA (IS11-8E10, FITC, Miltenyi Biotec), anti-IgM (MHM-88, FITC, BioLegend), anti-IgD (IA6-2, Alexa Fluor, BioLegend), anti-CD3 (UCHT1, FITC, BioLegend), and anti-CD14 (HCD14, FITC, BioLegend). Flow cytometric analysis and cell sorting was performed on a BD FACSJazz using the BD FACS Software. Plasmablasts were isolated by gating for CD19+CD20low/-CD27+CD38highIgA-IgM-IgD-CD3-CD14- cells. All paired chain antibody sequencing was carried out on IgG plasmablasts sorted into microtiter plates

at one cell per well by FACS and 1484 paired chain sequences were generated using Atreca's Immune Repertoire Capture® technology as previously described²⁷." (L358–L369)

Figure 1, advise to thin the gray line around boxes for visualization.

We have updated Fig. 1 according to the Reviewer's suggestion.

Line 132 – state briefly why 3 Abs were not tested (low expression?)

Two of the 18 Fabs that bound to Pfs25 monomer via ELISA did not show any detectable binding to Pfs25 when assessed via BLI, and the third mAb was not able to be expressed as a Fab. For additional clarification, we have added the following sentence to the manuscript. "Of the 18 mAbs predicted by ELISA to specifically interact with Pfs25, one mAb did not express in sufficient quantities as a Fab for analysis, and two mAbs did not bind Pfs25 with detectable affinity via BLI." (L133-135)

Line 144 – would it be useful to see if there is a correlation between affinities and oocyst reduction in SMFA? Apparently there is not.

In the manuscript, we report that there is no correlation between affinity and transmission reducing activity in SMFA. The associated p-value is not statistically significant ($p = 0.1231$).

Line 169 – a summary figure showing the different Mabs binding to their epitopes with their associated potency in SMFA will be nice.

To complement Fig. 7, we now include Fig. S8, which depicts the structures of the most potent mAbs from each epitope bin on Pfs25. The figure legend is as follows: "Structures of representative Fabs from each major epitope bin superposed on Pfs25. Site 1a (1269, green), Site 1b (2544, teal), Bridging (2586, pink), Site 2 (1245, salmon), and Site 3 (2530, orange), are labelled, and their associated affinity to recombinant Pfs25 and IC_{80} in SMFA are listed."

Line 243 – binding barely visible on the figure

We agree with the reviewer that the binding in Fig. 5, Panel F (gl2544+1) is difficult to see. However, we purposefully kept the same scale for the four kinetic curves in that panel to highlight the major differences a small amount of mutations can have on the kinetics of binding to recombinant Pfs25.

Figure 5 – will be nice to see germline gene VDJ and V-J as well in the alignment

To avoid overcrowding Fig. 5, we have added Fig. S6 to show a sequence alignment of the expressed 2544 mAb lineage members, their CDRs, and the position of their predicted V(D)J-encoding segments.

Line 282 maybe change “to” to “from” human vaccination

We have updated the text accordingly.

Discussion needs to be reviewed, the authors refer to HIV field, a lot has been hypothesized but not shown for HIV.

Structure-based vaccine design is an emerging strategy used across numerous infectious disease fields. At the Reviewer’s request, we have broadened the Discussion to now include examples beyond only the HIV field: “Such structure-based approaches have been successful in preferentially eliciting antibody responses to sites of vulnerability on the HIV Env trimer^{36,37}, the Respiratory Syncytial Virus (RSV) F protein³⁸ and the influenza Hemagglutinin^{39,40}, amongst other examples.” (L313-316) Several immunogens designed using structure-based approaches are currently being evaluated in various clinical trials (e.g. NCT03547245, NCT03699241, NCT03049488, NCT03814720, etc.) and the outcome of these experiments in human will reveal the full potential of this approach.

Reviewer #2 (Remarks to the Author):

This work expands our repertoire of human transmission blocking mAb as well as our understanding of the structure of the transmission-blocking epitopes present on Pfs25. The comparison with the germline mAb provides insight into the mutations required for high affinity binding that have the potential to inform the design of antigens that could be used sequentially to enhance the immune response. Although an attractive hypothesis for vaccine development, much further work is needed to test whether sequential antigens or another approach to focus the immune response toward specific epitopes will be sufficient to generate a robust transmission blocking response in humans. The lack of synergy between mAb that bound different epitopes was disappointing but important to know. Another challenge is the presence of antibodies that bind to the same region compete with one another yet have different transmission blocking potential. This issue should be included in the discussion.

We agree with the reviewer that the presence of antibodies targeting similar epitopes to 2544 could present an additional challenge through their partial competition with 2544, and have consequently added the following sentence to the Discussion: “Furthermore, with antibodies of less potency than 2544 partially competing for its epitope, it may be important to design an immunogen that preferentially elicits the most effective mode of binding to Pfs25.” (L318-320)

Prior to publication I think it is important to include additional details about the plasmablast sequence analysis. How many sequences were determined? It would also be good to include a phylogram of all the sequences with the selected Ig sequences (both those that did and did not bind Pfs25) and the location of the 2544 expansion indicated. This data would give a better overview of the response.

We have included additional sequence acquisition and sorting details in the Methods, as discussed above in response to Reviewer 1. We now also include a phylogram as Fig. S2, where antibodies and lineages are labelled as requested.

More detailed description of the structure of Pfs25 in relation to the EGF domains and the location of the gpi tail to indicate the potential orientation membrane as well as the proposed tiled alignment of Pvs25.

Because the monoclonal antibodies we report bind conformational epitopes, we elected to describe epitopes based on competition groups (Site 1, Site2, Site 3, bridging) as opposed to EGF domains. Nonetheless, we now include Fig. S5 that describes which EGF domains are contacted by the various antibodies for which a structure has been determined. In a previous report (Sally *et al.*, Nat Commun 2017) where we described the structures of Pfs25 bound to antibodies derived from Kymice, we discussed how the tiled-model previously proposed for Pvs25 is largely incompatible with our findings: “Saxena *et al.*¹³ had previously suggested a model that featured Pvs25 coating the cell surface of ookinetes in a tile-like fashion inferred from crystal packing interactions combined with residue conservation analyses. Our structural data shows that site 1-directed mAbs (e.g., 1269), which are active in SMFA, recognize a face of Ps25 that is buried and inaccessible at the interface of the Pvs25 tile-like model (Supplementary Fig. 11), arguing against the tile-like model for Pfs25. The epitopes of site 2-directed mAbs (e.g., 1245) are somewhat accessible in the tile-like model (Supplementary Fig. 11), but the fact that site 2 mAbs are less potent inhibitors than site 1 mAbs further argues against this arrangement of Pfs25 on the ookinetes surface.” We therefore prefer not to revisit this topic in the current manuscript. Nonetheless, we have modified Figs. 2, 3, and 7 to now include a label showing the position of the GPI anchor at the Pfs25 C terminus.

L114) They mention producing 38 Ig selected from most expanded lineages and then 15 fabs, but the methods indicates 96 sequences were selected for recombinant production and 75 were from the expanded lineages and the rest were highly mutated.

We thank the reviewer for their careful attention to the manuscript, and have corrected the text as follows: “A total of thirty-eight paired IgG variable regions (CDRs and framework regions) were selected for direct recombinant expression from the most expanded plasmablast lineages identified from the donor sample.” (L369-371)

Fig 2B) Please include the relative TRA and binding affinity.

We have included an additional column in panel B of Fig. S3, which describes antibody potency at 100 µg/mL. This Supplementary Figure also includes a thorough binding kinetics summary for the 15 Fabs that were initially expressed and characterized.

L153-169) Please add the location of the GPI tail to indicate the orientation with respect to the membrane and include the reference to the EGF domains targeted for all the Ig, since these are how the Pfs25 structure has been described historically.

As noted above, labels to the C terminus (indicating the position of the GPI anchor and the direct fusion to the VLP in the current immunogen) have been added to Figs. 2, 3, and 7. We unfortunately do not have a molecular definition of all IgG's, and can therefore not attribute the exact EGF domain(s) that constitute the epitopes. In fact, due to the Pfs25 fold, many of the antibodies bind across several EGF domains, and as stated above we believe it is more accurate to describe conformational epitopes by epitope bins, such as Site 1, Site 2, Site 3 and bridging. Nonetheless, we now include Fig. S5 that describes which EGF domains are contacted by the various antibodies for which a structure has been determined.

L177) Please include the TRA of mAb 4B7 as a comparator in this section.

Details regarding the TRA of mAb 4B7 have been included as follows: "Notably, 2544 has significantly higher TRA than any other mAb directed against Pfs25 yet reported, including the previously described 1269 Site 1-directed mAb derived from Kymice ($IC_{80} = 63 \mu\text{g/mL}$ [95% CI, 53–75]), and the murine 4B7 ($IC_{80} = 29 \mu\text{g/mL}$ [95% CI, 24–34])²⁷." (L 180)

Add the direct comparison of the VH and VL sequences of the 2544 Ig tested for transmission blocking activity in supplemental material using the format in Fig 5A.

As requested, we have added Fig. S7 to show an alignment of the V_H and V_K sequences of the reported 2544-lineage antibodies, their CDRs, and their predicted VDJ regions.

Reviewer #3 (Remarks to the Author):

This is a paper about trying to create a transmission blocking vaccine, something that is theoretically interesting but might be challenging in real life in humans. The authors have before published results from humanized mice, and have now moved on to humans which is very interesting, especially since the number of papers in the literature concerning trials in humans are not that big compared to the number of studies

performed in mice.

We thank the Reviewer for their interest and favorable assessment of our work.

To use only plasmablast analysis limits the interpretation of the studies, especially from a transmission blocking point of view where a long term response is needed to be efficient, but it is a good start towards understanding the basics of the immune response.

This study focuses on plasmablasts to concentrate our analysis on responses induced by the vaccine. It is our view that at this point it is important to know that a vaccine can produce effective lineages; this being the primary steps towards designing an effective vaccine. In this context, the primary output of vaccine-induced affinity maturation is the plasmablast. The induction of long-lived responses is very important as outlined by the Reviewer and will be the subject of subsequent studies. Those types of responses are difficult to profile, but plasmablast responses are a reasonable approximation (e.g. Sajadi *et al.*, Cell 2018).

Another limitation is that the plasmablasts have to survive for 7 days in culture, something that might in itself mean a selection bias.

Plasmablasts were not cultured, but were thawed, lysed, and sequenced all within a day. Therefore, lineages sequenced have proportional representation to the actual repertoire lineages that existed in the thawed PBMC sample's repertoire.

Some antibodies seem to partially impede access to important sites, hence it might be difficult to predict the total outcome of Pfs25 as a vaccine when as many as 4 EGF domains are included. EGF domains contain several S-S bridges and often fold well and form a relatively stable structure on their own, why have the authors not used a smaller protein with fewer EGF domains, why use 4 EGFs?

We agree with the Reviewer that using minimal domains of Pfs25 may be a useful approach to focusing the immune system to specific regions of Pfs25 that are of notable potency. However, this study represents the first molecular analysis of human plasmablast responses to Pfs25 immunization, which has helped elucidate these sites of vulnerability. Based on the findings described here, minimal domains of Pfs25 may be designed, but the characterization of such next-generation immunogens is beyond the current scope of the study.

To avoid forming (interfering) antibodies against too many sites, fewer EGF domains could potentially be used in a vaccine. It would therefore be good to include one figure to clarify which EGF domains are part of the different binding sites (site 1, site 2 etc) to facilitate an interpretation of the importance of the different domains. It would also be

good to test the mAb 2544 against the single EGF domains.

As per the above response to Reviewer 2, we now include Fig. S5 that describes which EGF domains are contacted by the various antibodies for which a structure has been determined. This analysis clearly shows how many of the antibodies bind across several EGF domains, including the most potent antibody we have identified, 2544, which binds across EGF 1, 3, and 4. As such, we believe it is more accurate to describe conformational epitopes by epitope bins, such as Site 1, Site 2, Site 3 and bridging. Although we believe minimal constructs can potentially be achieved, we envision designs that contain more complete conformational epitopes and have therefore not tested binding to individual EGF domains. Protein engineering approaches to achieve more efficacious Pfs25-based vaccines are ongoing and beyond the scope of the current study.

How do the authors explain that it is the heavy chain that seems to contribute mostly to the interface of the surface area of 2544 in the binding to Pfs25? Which subclass of IgG is it, does it make a difference?

Heavy chain residues being predominant in antibody-antigen interactions is considered relatively common, and therefore we do not think it is crucial to expand this point. All of the sequenced 2544-lineage mAbs were IgG1, with the exception of one mAb which was IgG2. While we agree with the reviewer that testing different classes of antibodies is potentially interesting, it is beyond the scope of this study given the majority of isolated antibodies were IgG1.

The authors state that they did not detect any relationship between mAb affinity to recombinant Pfs25 and potency in SMFA, and there was no association between binding strength and which epitope was targeted. Instead of looking only at KD values, was there any association with off-rates? They seem to differ more than on-rates, and off-rates should be important since if the antibodies only stay bound for a few seconds, they will not have time to confer any blocking effects.

The Reviewer raises an interesting point. However, when K_{off} 's reported in Fig. S3 are plotted against antibody TRA, there is still no clear relationship. To cover this point more broadly, we have updated the sentence in the Discussion to state: "Interestingly, we did not detect any relationship between mAb binding kinetics to recombinant Pfs25 and potency in SMFA." (L296-297)

The concentration of mAb needs to be in the range of 100 μ g/mL to have an effect on oocyst density. Is it realistic to think that this concentration can be achieved in human plasma? Are there other successful vaccines that have managed to reach this concentration of specific antibody?

We agree with the reviewer that 100 µg/mL is a high target to achieve and sustain by vaccination. The rationale of our approach, as stated in the Discussion is that “[...] eliciting high titers of the most potent mAb lineages will be critical to improve vaccine potency.” (L317-318) Therefore, the discovery of mAb 2544 with an IC₈₀ of 16 µg/mL allows us to postulate that if polyclonal responses with such a high potency can be preferentially elicited by next-generation immunogens, transmission reduction activity could occur at antibody titers achievable by vaccination. We also now note in the Discussion (L 335-338), “Combining next-generation immunogens with potent adjuvants that elicit high-titers and durable responses will also be critical for vaccine efficacy, as pre-clinical studies have shown how different combinations of Pfs25 and adjuvants can result in vastly different antibody responses⁴⁵⁻⁴⁷.”

The most expanded Pfs25-reactive plasmablast lineage was that of mAb 2544, and represented approximately 3% of the total repertoire, indicating that also other (potentially interfering) antibodies will be produced and that it will be difficult to reach a high concentration of the wanted antibody.

As described above in response to Reviewer 2, we agree that the presence of antibodies targeting similar epitopes to 2544 could present an additional challenge through their partial competition with 2544, and have consequently added the following sentence to the Discussion: “Furthermore, with antibodies of less potency than 2544 potentially competing for its epitope, it may be important to design an immunogen that preferentially elicits the most effective mode of binding to Pfs25.” (L318-320) We provide in the Discussion potential strategies for developing potent polyclonal responses, or even using passive immunization to circumvent this problem: “Combined, these data provide a guide for improved vaccine design based on immuno-focusing to mask immunogenic surfaces associated with poor mAb inhibitory activity and thereby direct B cell responses to sites associated with potent parasite inhibition. Such structure-based approaches have been successful in preferentially eliciting antibody responses to sites of vulnerability on the HIV-1 Env^{36,37}, the Respiratory Syncytial Virus (RSV) F protein³⁸ and the influenza hemagglutinin^{39,40}, amongst other examples.” (L311-316); and “Finally, our identification of a highly potent human mAb against Pfs25 offers the opportunity to test its efficacy in passive immunization trials, such as is currently being developed for other transmission-blocking targets e.g. Pfs48/45⁴²” (L339-341).

What is the actual concentration of the equivalent of 2544 in the human plasma, is it at all possible to measure it or detect it? If so, how long does it stay detectable in plasma? Since this paper has used only plasmablasts, it is of special interest to consider how long an immune response can be detected.

We appreciate these thoughtful comments from the Reviewer. We are not currently aware of a way to rapidly and quantitatively analyze levels of a specific mAb over time, or mAb lineages in human plasma. Methods that use proteomic approaches

combined with sequence matching from plasmablast deep sequencing analysis are being pioneered for some vaccine evaluation (e.g. Wine *et al.* *Curr Opin Immunol*). While this may be of interest in the future, it is outside the scope of this study. In addition, access to samples is limited for this clinical study that prevents a more in-depth analysis of B cell subsets that would inform on the durability of the response.

In the discussion about the number of mutations from a germline B cell precursor, it would be of interest not only to note the number of amino acid mutations but also how big a structural change every mutation means. Some amino acids are very similar in overall structure/side chains and a change for another amino acid might not mean a big change in structure.

As stated in L209-212: “To understand the structural basis of affinity maturation, we solved the crystal structures of the unliganded 2544 Fab and unliganded g12544 Fab at 2.4 Å and 2.0 Å resolution, respectively (Table S1).” In addition, on L226-230: “From our comparative structural studies, we propose that there is an interconnectedness between the light chain N terminus, κCDR1 and κCDR3 where somatic hypermutations in the κCDR3 result in stabilizing light chain components in an optimal configuration for engaging Pfs25, from an otherwise flexible conformational ensemble in the germline antibody (Fig. 5C).” Individual amino acids that we postulate have evolved towards shaping the paratope in a conformation optimal for Pfs25 recognition are highlighted in Fig. 5E. The contribution of these individual mutations is supported by binding data using reversion mutants of the antibody presented in Fig. 5D. The top five mutations identified have amino acid exchanges of considerable differences in terms of side chain properties (e.g. P->F; Y->D; S->G; T->P; A->T).

Reviewers' Comments:

Reviewer #1:

Remarks to the Author:

Comments have been addressed in this revised version of the manuscript.

Reviewer #2:

Remarks to the Author:

The revised manuscript seems fine. I appreciate that addition of the C-term label and the plasmablast section of the methods is clear. However, I think the wrong figure is referenced on Line 150. I do not think it should be Fig S1. Hope the future targeted immunization strategy goes well.

Reviewer #3:

None

REVIEWERS' COMMENTS:

Reviewer #1 (Remarks to the Author):

Comments have been addressed in this revised version of the manuscript.

We thank the reviewer for their time and expertise.

Reviewer #2 (Remarks to the Author):

The revised manuscript seems fine. A appreciate that addition of the C-term label and the plasmablast section of the methods is clear. However, I think the wrong figure is referenced on Line 150. I do not think it should be Fig S1. Hope the future targeted immunization strategy goes well.

We thank the reviewer for their careful attention and encouraging comment; the correct figure (Supplementary Figure 3) is now referenced.